# Electrically driven amplification of terahertz acoustic waves in graphene

Aaron H. Barajas-Aguilar [1], Jasen Zion[2], Ian Sequeira[1], Andrew Z. Barabas[1], Takashi Taniguchi [3], Kenji Watanabe [3], Eric B. Barrett[1], Thomas Scaffidi[1] & Javier D. Sanchez-Yamagishi [1] ✉

In graphene devices, the electronic drift velocity can easily exceed the speed of sound in the material at moderate current biases. Under these conditions, the electronic system can efficiently amplify acoustic phonons, leading to an exponential growth of sound waves in the direction of the carrier flow. Here, we show that such phonon amplification can significantly modify the electrical properties of graphene devices. We observe a superlinear growth of the resistivity in the direction of the carrier flow when the drift velocity exceeds the speed of sound – resulting in a sevenfold increase over a distance of 8 μm. The resistivity growth is observed at carrier densities away from the Dirac point and is enhanced at cryogenic temperatures. We develop a theoretical model for the resistivity growth due to the electrical amplification of acoustic phonons – reaching frequencies up to 2.2 THz – where the wavelength is controlled by gate-tunable transitions across the Fermi surface. These findings provide a route to on-chip high-frequency sound generation and detection in the THz frequency range.

Sound waves are important as high-frequency signal carriers and as a means to distort crystal lattices in space and time. Due to the slow speed of sound compared to light, ultrashort sound wavelengths in the nanometer scale are attainable in the terahertz (THz) domain, the highest sound frequencies in solids. The control and generation of THz sound waves offers a route to nanoscale ultrasonic imaging, the generation of THz electromagnetic radiation[1-3], and the dynamic modulation of electronic behaviors[4-7]. However, an electrical on-chip source of THz acoustic waves remains elusive. Coherent THz sound waves have only been achieved via ultrafast optical pumping[8,9], while conventional piezoelectric transducers produce maximum frequencies of ~10 GHz[10].

Accelerated electrons can generate and amplify traveling waves. Famous examples include the traveling wave amplifier[11] and Cherenkov radiation. The common gain condition is that the electron velocity exceeds the wave phase velocity. In solids, an analogous effect occurs when the carrier drift velocity ($v_D$) exceeds the speed of sound ($v_S$), resulting in acoustic wave amplification[12,13]. Unlike the free

electron case, the amplification has a characteristic wavelength given by transitions across the Fermi surface (Fig. 1a). Such acoustic amplification has been studied in bulk semiconductors[14,15] and has recently been used to make nonreciprocal acoustic amplifiers operating in the gigahertz frequency range[16].

Two-dimensional van der Waals (vdW) materials present many advantages for acoustoelectric devices because phonons are naturally confined to atomic layers, leading to long lifetimes[17] and more efficient coupling to electrons[18]. Acoustic waves offer a way to dynamically modulate lattice strain in both space and time, which can couple to diverse vdW heterostructure phenomena[19], as well as to quantum defects[20]. Despite this interest, acoustic studies in vdW materials have been limited by the challenge of generating propagating sound waves at high frequencies, as the confined nature of the two-dimensional (2D) phonons makes it difficult to excite with external transducers.

Graphene is an attractive host for the electrical amplification of acoustic phonons. Its high Fermi velocity and electron mobility ($\mu$)

[1]Department of Physics and Astronomy, University of California, Irvine, Irvine, CA, USA. [2]T. J. Watson Laboratory of Applied Physics, California Institute of Technology, Pasadena, CA, USA. [3]Research Center for Functional Materials, National Institute for Materials Science, 1-1 Namiki, Tsukuba, Japan. ✉e-mail: javier.sanchezyamagishi@uci.edu

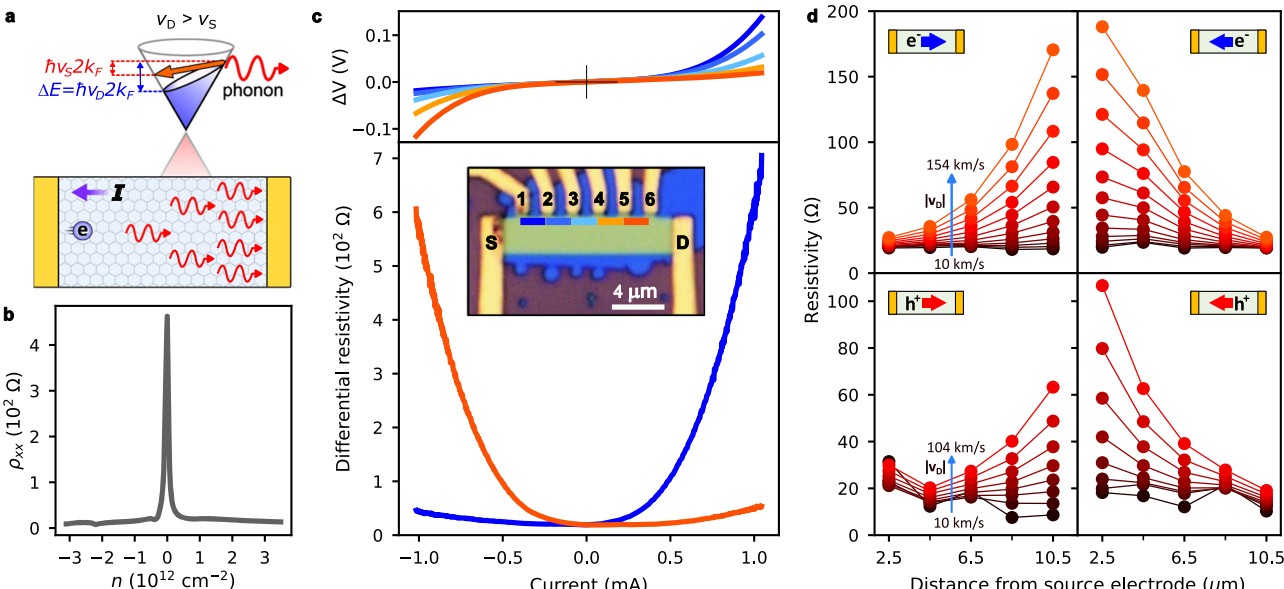

**Fig. 1 | Acoustic phonon amplification and observation of resistance growth in the direction of carrier flow. a** Top: Schematic of the graphene electronic distribution with a drift velocity ($v_D$). The blue shaded region shows the occupied states. The Fermi surface is tilted such that the energy difference between right and left moving carriers is $\hbar v_D 2k_F$. When $v_D > v_S$, the carriers can backscatter and emit a phonon of energy $\hbar v_S 2k_F$ (transition is indicated by the orange arrow). Bottom: device under the phonon amplification conditions, the phonon population grows exponentially with distance in the direction of the carrier flow. **b** Resistivity versus electron carrier density for device A, the absence of satellite peaks indicates that the graphene and hBN layers of the device are unaligned. **c** top: Voltage difference ($\Delta V$) between pairs of consecutive contacts vs. source-drain current for device A. The voltage differences $\Delta V_{1-2}$ (blue) and $\Delta V_{5-6}$ (orange) exhibit the largest non-Ohmic behavior ($n = 1.4 \times 10^{12}$ cm$^{-2}$), bottom: differential resistivity vs. source-drain

current for the outermost pairs of contacts ($\Delta V_{1-2}$ and $\Delta V_{5-6}$) showing asymmetric non-Ohmic behavior. Inset: the optical image of device A with a 13 μm length, 3 μm width, and center-to-center distance between voltage probe contacts of 2 μm. The colored bars label the pairs of contacts used to measure the voltage differences plotted in the top and bottom panels. **d** Position dependence of the resistivity at different drift velocities for $n = +1.4 \times 10^{12}$ cm$^{-2}$ (top panels) and $n = -1.4 \times 10^{12}$ cm$^{-2}$ (bottom panels). Lines are a visual aid connecting data points. The device cartoons show the carrier flow direction and carrier type for each case, the source contact is on the left, as in (**c**). The maximum drift velocity in the top panels is 154 km/s ($j = 0.34$ mA/μm), and 104 km/s in the bottom panels. Note that the maximum $v_D$ achieved for holes is lower than for electrons due to larger source-drain contact resistance. All measurements are performed at $T = 1.5$ K.

means that large drift velocities ($v_D = \mu E$) can be achieved at relatively small electric biases. Under a current bias, the drifting Fermi distribution results in an effective population inversion with an energy difference $\Delta E = \hbar v_D 2k_F$ between forward and backward propagating carriers (at zero temperature), where $k_F$ is the magnitude of the Fermi wavevector. When $v_D > v_S$ ($v_{S-TA} = 13$ km/s and $v_{S-LA} = 21$ km/s for TA and LA phonons, respectively[21]), electrons can relax energy and momentum by emitting and amplifying waves via inelastic backscattering (Fig. 1a). Importantly, over a large range of parameters, the only excitations that graphene can generate are acoustic waves[22,23], which are in the THz frequency range (Supplementary Fig. 6). Evidence for acoustic phonon amplification has come from noise measurements in graphene devices[24]. However, the direct effects of THz acoustic waves on the electronic properties of materials are still unexplored.

Here, we study the transport behavior of clean graphene devices as a function of voltage bias at cryogenic temperatures. In contrast to previous bias studies[25–28], we measure the position-dependent resistance, focus on gate voltages away from the Dirac point so interband transitions are suppressed, and apply moderate source-drain voltage biases to avoid optical phonon generation. Our primary findings are that graphene resistivity grows strongly in the direction of carrier flow when the drift velocity exceeds the speed of sound. Our results are well explained by the electrically-induced amplification of terahertz acoustic waves and their associated strong scattering of graphene electrons.

## Results

To study the spatial dependence of the graphene resistance, we fabricated long graphene devices encapsulated in hexagonal boron

nitride (hBN) with equally spaced voltage probes along the channel length (inset Fig. 1c). A DC source-drain voltage bias is applied to the device, and by measuring the voltage difference between adjacent electrodes we probe the spatial distribution of the voltage drop across the channel as a function of the current flowing between the source and drain electrodes. The resulting V–I curves are shown in Fig. 1c (top panel), measured at an electron carrier density $n = 1.4 \times 10^{12}$ cm$^{-2}$ and cryostat temperature of 1.5 K. At low currents, all curves show linear Ohmic behavior corresponding to an average resistivity of 19.7 Ω/square with less than 6% deviations across the sample. The average carrier mobility is $2.3 \times 10^5$ cm$^2$/V*s. At higher current magnitudes, all the curves deviate strongly from Ohmic behavior with a differential resistance (d$V$/d$I$) that grows superlinearly with the magnitude of the source-drain current (Fig. 1c, bottom panel).

Strikingly, the nonlinear resistance is highly asymmetric with the current direction and strongly position-dependent. The largest asymmetry and nonlinearity are found for the measurements closest to the source and drain electrodes (blue and orange lines Fig. 1c). For example, for contacts 1–2, the differential resistivity at −1 mA is 44 Ω, but at +1 mA it rises to 612 Ω, a factor of 14× difference for opposite current directions. Measurements on the opposite side of the device (5–6) produce similar nonlinear curves but with opposite dependence on the current direction.

The position dependence of the graphene resistivity shows superlinear growth in the direction of carrier flow (Fig. 1d): For electron-doping, the resistivity growth is opposite to the current; for hole-doping, growth is in the direction of the current. The growth is substantial at moderate current densities $j = 0.34$ mA/μm, leading to a 7× increase in resistivity and 12× increase in differential resistivity over

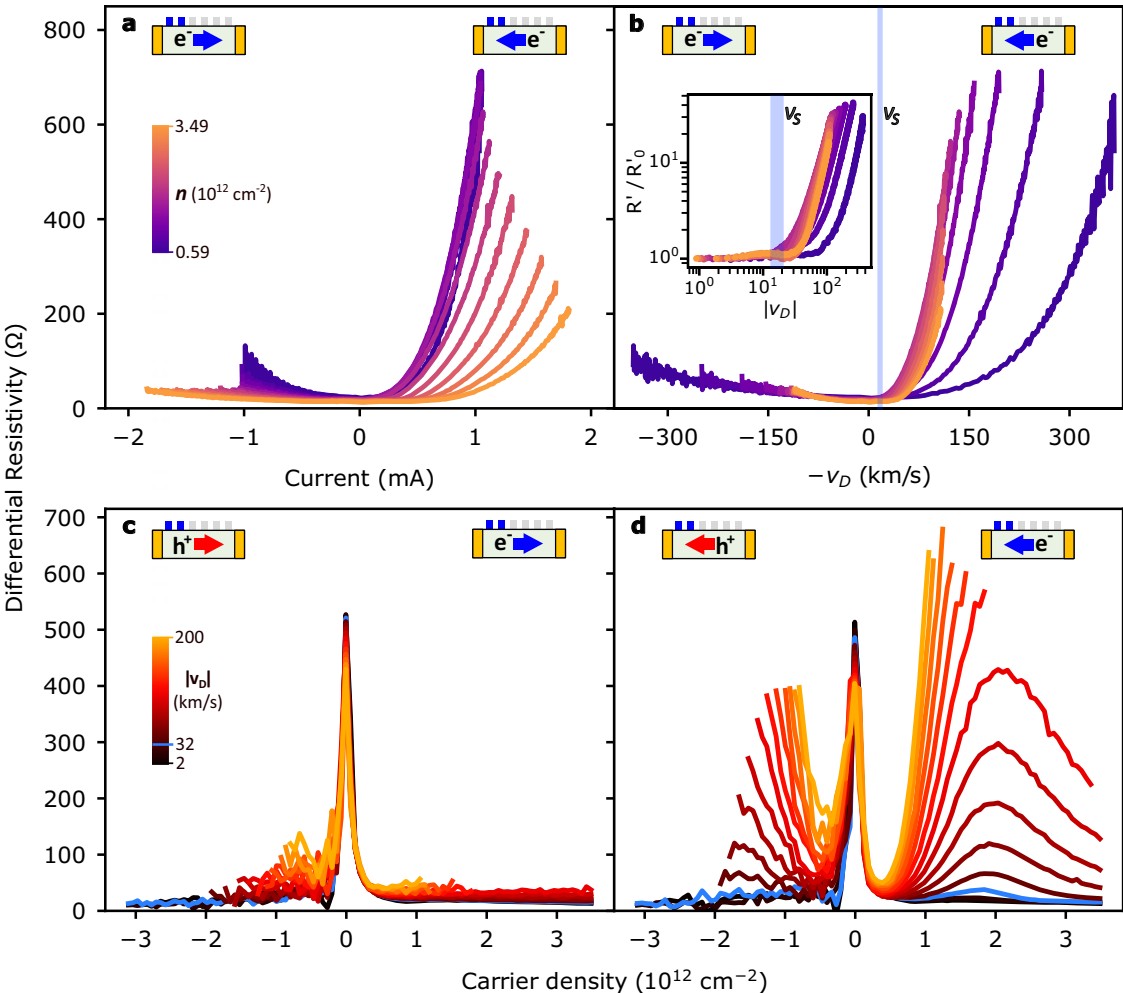

**Fig. 2 | The resistance growth is sensitive to carrier density and only occurs when $v_D > v_S$.** Differential resistivity as a function of current (**a**) and drift velocity (**b**) at different carrier densities. The shadowed regions indicate drift velocities between 13 and 21 km/s, corresponding to the speed of sound for TA and LA phonons, respectively. Inset: Logarithmic plot of differential resistivity normalized to the value at $v_D = 0$. **c**, **d** Same differential resistivity data plotted vs. carrier density for constant drift velocities (positive and negative for (**c**) and (**d**), respectively). The device cartoons indicate the carrier type and flow direction, as well as the contacts being measured in each case (contacts 1–2 for all the panels). The highlighted blue traces correspond to a drift velocity of 32 km/s, which is the lowest drift velocity above $v_{S-LA} = 21$ km/s shown in this plot. All measurements are performed at $T = 1.5$ K. Note that data were taken within a maximum source-drain voltage of $\pm 0.6$ V. Hence the curves appear with different ranges when plotted versus current, $v_D$, or for constant values of $v_D$.

8 μm (Supplementary Fig. 1). As a result of the highly nonuniform resistivity at this bias, 81% of the graphene resistance occurs in the last 46% of the channel length (Supplementary Fig. 2).

The nonlinear resistance growth occurs for a wide range of carrier densities. This is seen in Fig. 2a, where, for contacts 1–2, we observe strongly asymmetric d$V$/d$I$ versus current curves for carrier densities ranging from $0.6 \times 10^{12}$ to $3.5 \times 10^{12}$ cm$^{-2}$. For all densities, the resistance is larger in the direction where the carriers travel longer distances in the device (downstream, Fig. 2a).

When plotting d$V$/d$I$ versus drift velocity ($v_D = j/ne$) (Fig. 2b), the curves with carrier densities above $1.1 \times 10^{12}$ cm$^{-2}$ collapse together, suggesting that the physics of this phenomenon is dictated by a drifting electronic carrier distribution. On a logarithmic scale plot (Fig. 2b inset), the normalized differential resistivity versus $v_D$ shows a sharp transition from constant to a growing non-Ohmic behavior. The threshold drift velocities, which we define as a 1.5× increase in differential resistivity, vary with carrier density from 21.8 to 83.3 km/s. Notably, non-Ohmic behavior is only observed above the lowest graphene sound velocity ($v_{S-TA} = 13$ km/s). The sharp transition between the Ohmic and non-Ohmic regimes can be observed for any pair of contacts in the device (Supplementary Fig. 3).

When the voltage drop is measured closest to the carrier injection point (upstream, Fig. 2c), the resistance versus carrier density curves follow the typical graphene Dirac peak response with a weak dependence on drift velocity, indicating mostly Ohmic behavior. When the carrier flow is reversed, the carriers travel 10.5 μm before reaching the measuring contacts (downstream, Fig. 2d). Here, for drift velocities lower than $v_S$, the line traces show a typical graphene response–decreasing in resistance with increasing carrier density magnitude (Fig. 1b). But, when $v_D$ is larger than the speed of sound (light blue line), the differential resistivity instead grows rapidly with carrier density for $n$ larger than $0.4 \times 10^{12}$ cm$^{-2}$, surpassing the value of the resistance at the Dirac peak. This effect can be seen symmetrically for both electron and hole carriers. As the carrier density increases, we observe a peak in the d$V$/d$I$ at $n \sim 2 \times 10^{12}$ cm$^{-2}$ and an eventual downturn at higher electron doping. Similar behavior is observed for all the other pairs of contacts along the device (Supplementary Fig. 4).

The resistance growth is most prominent at cryogenic temperatures. Figure 3a displays the differential resistivity versus drift velocity measured for contacts 1–2 from $T = 1.5$ to 280 K. At $T = 1.5$ K, a highly asymmetric curve is obtained with a large nonlinear resistivity growth when measuring far from the carrier injection point (downstream,

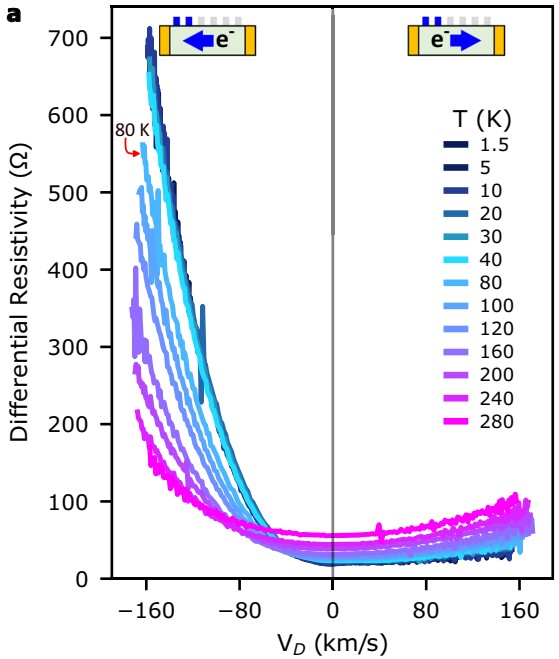

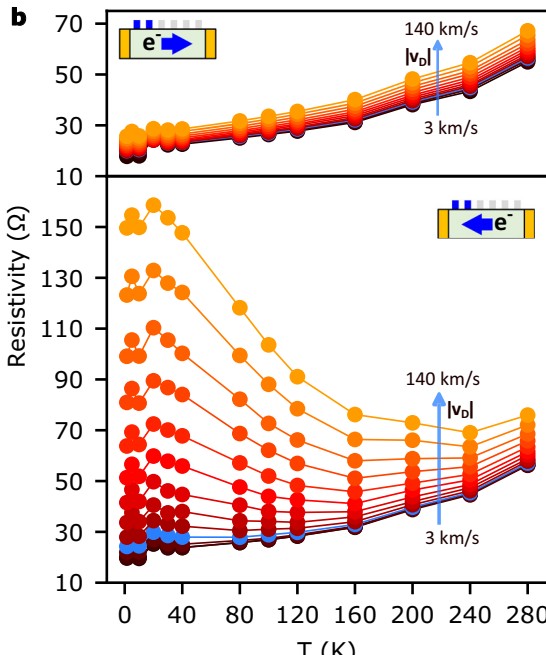

**Fig. 3 | Resistance due to amplified acoustic phonons is largest at low temperatures. a** Differential resistivity vs. drift velocity was measured for contacts 1–2 at different temperatures from 1.5 to 280 K at $n = 1.4 \times 10^{12}$ cm$^{-2}$. **b** Resistivity for contacts 1–2 as a function of temperature for constant drift velocities. The blue arrows indicate the direction of growth of the $v_D$ magnitude. The blue traces correspond to $v_D = 37$ km/s, from which the temperature dependence of the resistivity inverts when the carriers move downstream. The excess resistivity induced by high $v_D$ at 1.5 K is 130 Ω. The excess resistivity induced by heating from 1.5 K to 280 K for $v_D = 3$ km/s is 37 Ω.

negative $v_D$). As the temperature increases, the nonlinearity and asymmetry are steadily reduced but are still evident even at $T = 280$ K. Correspondingly, the resistance dependence on temperature shows an opposite trend for different current bias directions (Fig. 3b). In an upstream configuration (top panel Fig. 3b), a steady increase of the resistivity with temperature is observed as is typical for graphene at low biases, consistent with increased scattering from thermally-occupied acoustic phonons[29,30]. For the downstream measurement (bottom panel Fig. 3b), a similar behavior is observed for small drift velocities. But, as $v_D$ increases beyond 37 km/s (blue trace), the opposite dependence is observed, with resistivity decreasing as the temperature rises above 40 K. At 1.5 K, the effect of drift velocity on resistivity is 3.5 times greater than the effect of heating from 1.5 K to 280 K.

To summarize, we observe the graphene resistivity to grow superlinearly in the direction of carrier flow when the drift velocity exceeds the speed of sound. The resistance growth is suppressed at low carrier densities, and it is strongest at cryogenic temperatures. These observations were made in two graphene devices where the graphene is misaligned with its encapsulating hBN layers (Supplementary Figs. 8–10). The directional resistance growth is not observed in devices where the graphene and hBN form a moiré superlattice (Supplementary Fig. 11) or where there is too much disorder (Supplementary Fig. 12).

## Discussion

In low-disorder conductors, acoustic phonons are the simplest low-energy excitation that can relax the momentum of electrons accelerated by electric fields and create resistance. At low temperatures, this form of dissipation is only unlocked when the drift velocity exceeds the speed of sound due to energy-momentum conservation. This can be understood from the form of the tilted Fermi distribution in a drifting rest frame $f_D(\epsilon) = 1/(1 + \exp(\epsilon - \mu - \hbar v_D \cdot k)/k_B T)$, where the effective population inversion between forward and backward moving carriers enables electrons to backscatter by emitting acoustic phonons

in the direction of the carrier flow, with characteristic wavevectors $k_{phonon} \sim 2k_F$ and energy $\hbar v_S k_{phonon}$ (Fig. 1a).

In this work, the range of drift velocities where the resistivity growth is observed points to an acoustic-phonon mechanism. The threshold behavior for the drift velocity is particularly sharp, with little change to the local resistivity (<15%) and highly symmetric V-I curves until the sound velocity is exceeded (Fig. 2b). When $v_D = v_{S-LA}$, the tilt of the Fermi distribution is $\hbar v_D 2k_F = 5.8$ meV at $n = 1.4 \times 10^{12}$ cm$^{-2}$. At these low energy scales, acoustic phonons are the only excitation available for inelastic energy transfer. At the highest drift velocities that we probe at this carrier density (200 km/s), the energy tilt is 55 meV and is not sufficient to directly excite the lowest energy optical phonons of the device (102 meV for hexagonal boron nitride[31]).

Typically, electron-phonon scattering is considered solely a source of dissipation, leading to local heating of the electron and phonon distributions. However such Joule heating effects would not produce the strongly asymmetric resistance profile that we observe (see Methods section on heating effects). Instead, if the acoustic phonons are long-lived, they will propagate downstream and stimulate the emission of additional phonons, producing an exponential growth in the direction of carrier flow. The only condition for growth is that the net rate of phonon emission exceeds the phonon decay rate, the latter being small at cryogenic temperatures[32]. The phonon population growth will be mirrored in the resistivity, as each phonon emission occurs with an electron backscattering. Thus, acoustic-phonon amplification results in an exponentially growing resistance in the direction of carrier flow when $v_D > v_S$.

We calculate the phonon amplification rates and effects on the graphene resistance using a model of the driven electron–phonon dynamics across the channel length (see Methods and SI Section 5). We find a cone of phonon modes in the direction of the drift velocity that will be amplified, resulting in exponential growth in the direction of the carrier flow as $\exp(\Gamma_k^*(x/v_s))$, where $\Gamma_k$ is the amplification rate for mode $k$, and $x$ is the position. $\Gamma_k$ reaches levels of 1–40 GHz for $v_D$

values from 2 to $10*v_{S-LA}$, with a broad maximum near $k = k_F$ along the $x$ direction (Supplementary Fig. 13). Extending beyond previous works, we calculate the spatial growth rate of the resistivity, and find it to be well approximated by the peak value of $\Gamma_k/v_S$ with values of 0.3–2.5 $\mu m^{-1}$ (Supplementary Fig. 14). Such micron-scale growth lengths are comparable with our experimental observations. For the data measured at $v_D = 7.33*v_{S-LA}$ (154 km/s), $n = 1.4 \times 10^{12}$ $cm^{-2}$, the observed trend is well fit by an exponential with a characteristic growth rate ~0.32 $\mu m^{-1}$ (Supplementary Fig. 2). This value is 5× less than the theoretical calculation, which is not surprising given that the model neglects phonon loss mechanisms such as anharmonic decay and edge scattering.

The nonmonotonic dependence of the resistivity growth on carrier density can also be understood within the phonon amplification model. Near the Dirac point, we expect phonon amplification to be suppressed due to competing pathways for energy relaxation via interband excitations, such as electron–hole generation, when the Fermi tilt is comparable to the Fermi energy[27]. Moreover, the electron–phonon coupling and amplification rates are reduced with smaller $k_{phonon} \sim k_F = \sqrt{\pi n}$ (Supplementary Fig. 13). This explains why at lower carrier densities, higher drift velocities are required to observe the phonon amplification effect (Fig. 2b inset). Conversely, as the carrier density increases, the larger $k_{phonon}$ will have stronger electron–phonon couplings and subsequent higher rates of amplification and resistive scattering with electrons[24,29]. The larger Fermi surface also amplifies a larger range of modes, further increasing the resistivity growth. An opposing effect is the increase of the phonon decay rates at larger phonon wavevectors[33], either due to anharmonic decay or short-range disorder, which will lower the amplification rates at higher carrier densities. Our data is in agreement with these aspects of the phonon amplification model, where we observe a sharp increase in the resistivity growth as we dope away from the Dirac point, a maximum at $n \sim 2 \times 10^{12}$ $cm^{-2}$, and a downturn for higher $n$ values (Fig. 2d).

The requirement that the phonon emission rate exceeds the decay rate for net amplification makes the process also sensitive to the overall lattice temperature. The phonon decay rate will increase with thermal mode occupation due to anharmonic processes, which will suppress the amplification process. Indeed, the resistance growth is strongest at $T = 1.5$ K and is only suppressed when the temperature exceeds the energy scale of the amplified modes ($\hbar v_S 2k_F/k_B = 40$ K for $n = 1.5 \times 10^{12}$ $cm^{-2}$) (Fig. 3b). The unique end result is a metallic conductor with a larger resistance at cryogenic temperatures than at room temperature when biased.

Other processes that scatter acoustic phonons will also suppress phonon amplification. This explains why we do not observe the resistance growth in disordered devices (i.e., with low electronic mobility, Supplementary Fig. 12) or with a moiré superlattice potential (Supplementary Fig. 11). In the case of aligned graphene-hBN devices, the moiré superlattice can be as large as 14 nm, which is comparable to the wavelengths of the emitted phonons (9–40 nm). As such, the superlattice will induce phonon–phonon Umklapp scattering, which will reduce the phonon lifetime and the amplification effect[34].

In summary, we demonstrate an unprecedented directional growth of the graphene resistance induced by electrically amplified acoustic phonons. Our results have important implications for graphene applications, especially at high current densities or over long distances where phonon amplification is likely to be a limiting factor. At the same time, these observations show the unique potential of high-frequency acoustic waves to remotely modulate the electrical properties of a material. The strong modulation of the graphene resistance is only possible due to the large wavevectors of the generated phonons, which can backscatter electrons across the Fermi surface. Low wavevector phonons with $k_{phonon} \ll 2k_F$, as would be predominantly generated by a thermal pulse, can only induce small-angle scattering and hence weakly affect the resistivity. For similar reasons, traditional acoustic-electronic

studies, which use surface acoustic waves with wavelengths ≫ 100 nm, can only act as slow and long length-scale perturbations for carrier densities above $1 \times 10^{11}$ $cm^{-2}$ ($\lambda_F < $ ~100 nm). In our devices, we estimate that we are predominantly amplifying acoustic phonons with wavelengths of 40 nm down to 9 nm. Interestingly, such length scales are comparable to the moiré superlattices found in twisted van der Waals heterostructures[35,36], motivating future work studying the dynamic spatiotemporal strain effects of acoustic waves.

The characteristic frequencies of the amplified acoustic phonons are in the terahertz range (0.3–2.2 THz) and are tunable with the graphene Fermi energy $E_{phonon} \sim (v_S/v_F)E_F$ (Supplementary Fig. 6). Currently, there are no alternative demonstrations of electrical generation of terahertz acoustic phonons. Transducing the mechanical motion of the acoustic wave to an electric field would offer a route toward a terahertz electromagnetic source. Acoustic amplification should be observable in other high-mobility vdW materials, such as transition metal dichalcogenides[37], where intrinsic piezoelectricity can convert the sound waves into electric waves. Lastly, Cherenkov phonon amplification in graphene offers a unique route to the electrical generation of other high-frequency and large wavevector excitations in 2D heterostructures, such as acoustic plasmons[38] and magnons[39], which are otherwise challenging to source and probe.

## Methods

### Fabrication

All graphene and hBN layers were exfoliated from bulk crystals. Stacks were fabricated by the dry transfer method[40] using stamps made of a polycarbonate (PC) film on top of a polydimethylsiloxane (PDMS) square on a glass slide. All the lithographic processes were made by electron beam lithography (EBL) using a layer of poly(methyl methacrylate) (PMMA) resist. To write the patterns for the one-dimensional (1D) edge contacts, PMMA 950 A5 was spun for 2 min at 2000 rpm producing a ~500 nm thick layer. The EBL patterns were written at 1.6 nA with 30 kV excitation and then developed for 2–3 min in a cold mixture of 3:1 isopropyl alcohol (IPA)/water. After writing and developing the patterns, reactive ion etching (RIE) was used to expose the graphene with the following parameters: a flow of 10 standard cubic centimeters per minute (SCCM) of $SF_6$, 2 SCCM of $O_2$, and 30 W of radio frequency power, at 100 mTorr for 30 s[41]. Then, 3 nm of Cr and 100 nm of Au were deposited in an electron beam evaporator system at 1 Å/s. Liftoff was performed by soaking the sample in acetone for 1–2 h and agitating with a pipette. To define the geometry of the device, a mask was written with EBL, and finally, a two-step RIE process was made, first an $SF_6$ etching using the same parameters described for the 1D edge contacts and then an $O_2$ etching with a flow of 20 SCCM of $O_2$, 30 W of radiofrequency power at 70 mTorr for 15 s.

### Device measurements details

The devices were measured in a variable temperature cryostat. For the transport measurements, a source-drain DC voltage bias was applied to the devices while measuring the sourced current using a Keithley 2400 SMU. The voltage drop between consecutive pairs of contacts was measured using digital multimeters. A gate voltage ($V_G$) was applied to the silicon back gate to control the carrier density. To calibrate the gate capacitance value for device A in the main text, which determines the calculated values of the drift velocity, the Landau fan features up to $B = 3$ T were measured and fitted. Using the measured capacitance and thicknesses of the dielectric layers, we extract a value for the out-of-plane dielectric constant of hBN $\varepsilon_\parallel = 3.44$, which agrees with the reported value[42]. This dielectric constant value for hBN was used in the data analysis for the rest of the devices.

### Data analysis

From the measurements, we obtain a dataset of $V$-$I$ curves for each pair of consecutive contacts over a range of applied gate voltages (−50 to

50 V). From these data, we calculate the local differential resistivity and local resistivity between adjacent contacts, which we use to plot the spatial resistivity profile of the device (Fig. 1c bottom and Fig. 1d). We also calculate the carrier density $n = CV_G/e$ (where $C$ is the gate capacitance per area and $e$ is the electron charge) and the drift velocity $v_D = j/ne$ for each data point, which is used to create Fig. 2. To produce curves of constant $v_D$ values, we use 1D interpolation to determine the differential resistance for equally-spaced $v_D$ values (Fig. 2c, d). This procedure is well-behaved, as the differential resistance is a smooth function of $v_D$ (Fig. 2b).

### Uncertainty analysis

The maximum uncertainties for measurements of the current and voltages are 0.1% and 0.2%, respectively. This corresponds to a maximum uncertainty of 0.2% for the resistance and differential resistance. Considering a 100 nm accuracy in the dimensions of the device, the uncertainty for the resistivity and differential resistivity is 6%. The uncertainty in the carrier density is determined by the 5% uncertainty in the dielectric constants of the hBN and $SiO_2$. This corresponds to a maximum uncertainty of 6% for the carrier density and 7% for the drift velocity. In summary, none of these uncertainties affect the conclusions reached in this work.

### Theory calculations

To calculate the resistance in a long graphene channel under a current bias, we assume a drifting Fermi–Dirac distribution with drift velocity $v_D$ for the electrons in order to calculate the phonon amplification rate $\Gamma_q^{amp}$ due to electron-phonon coupling via the deformation potential. We then use it to calculate the position-dependent out-of-equilibrium phonon distribution in the sample. Finally, we find the position-dependent electric field needed to sustain the assumed $v_D$ based on the electronic Boltzmann equation in which the phonon distribution enters through the electron-phonon scattering integral.

Our primary assumption is that of a drifting Fermi–Dirac distribution for the electrons, with a nominal temperature that is $v_D$-independent. In the actual system, we would expect that Joule heating would lead to increased electronic temperatures at high $v_D$. We also neglect phonon loss mechanisms such as disorder scattering or anharmonic decay, which will limit the phonon amplification process at large temperatures and large phonon population levels. As such, the model is best at capturing the qualitative aspects of phonon amplification, especially at lower drift velocities. For a detailed explanation of the theoretical calculations, see Supplementary Note 5.

### Relation to previous noise studies of acoustic-phonon amplification in graphene

A previous work by Andersen et al.[24] provided evidence for acoustic-phonon amplification in graphene via noise measurements and by extracting phonon transit timescales from AC transport measurements. Our study is differentiated by showing how acoustic-phonon amplification can directly modify the graphene DC resistance, leading to dramatic and easily measurable effects that are spatially varying.

### Other mechanisms for spatially-dependent resistance at high bias

Large source-drain voltage biases can lead to a spatially dependent carrier density across the channel[26]. This can lead to a spatially varying device resistance which is significant if the voltage bias is comparable to the gate voltage. In our measurements, the maximum applied biases are 0.6 V, leading to a $\Delta n \sim 4 \times 10^{10}$ cm$^{-2}$ change in the carrier density across the device channel. This is small (1%) compared to the much larger gate voltages and carrier densities, which are the focus of the presented data ($V_G = \pm 50$ V and $n \sim \pm 3.3 \times 10^{12}$ cm$^{-2}$), and thus cannot explain the observed resistance growth (Supplementary Fig. 7).

Joule heating due to resistive dissipation can lead to a spatially-dependent temperature profile with corresponding spatially-dependent resistances and thermovoltages. In a diffusive system, Joule heating depends only on the magnitude of the current, leading to a temperature profile that is symmetric with the current. The long devices that we measure are in the diffusive regime, as the electron mean free path is substantially smaller than the channel length due to edge scattering (sample width = 3 μm, length = 13 μm) and will get smaller as the electron-acoustic phonon scattering increases when $v_D > v_S$. Thus, in our devices, Joule heating is expected to produce a symmetric temperature profile and cannot explain the strongly asymmetric resistance profile we observe.

## Data availability

The data for device A appearing in the main text of this study have been deposited in the Zenodo database. Additional data for devices A–D are provided in the Supplementary Information file.

## Code availability

The code for the experimental data analysis is available from the corresponding author upon request. The code for the theoretical calculations is available at GitHub.

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

## Acknowledgements

The authors acknowledge the use of facilities and instrumentation at the Integrated Nanosystems Research Facility (INRF), in the Samueli School of Engineering at the University of California, Irvine, and at the UC Irvine Materials Research Institute (IMRI), which is supported in part by the NSF MRSEC through the UC Irvine Center for Complex and Active Materials. The authors also acknowledge the use of the UCI Laser Spectroscopy Lab. The authors thank L. Jauregui, V. Fatemi, and E. Pop for productive discussions, as well as the technical assistance of Q. Lin, R. Chang, M. Kebali, J. Hes, and D. Fishman. This work was partially supported by the National Science Foundation Career Award DMR-2046849. A.H.B.A. acknowledges the University of California Institute for Mexico and the United States (UC MEXUS) for partial financial support. I.S. acknowledges fellowship support from the UCI Eddleman Quantum Institute.

## Author contributions

A.H.B.A., J.Z., I.S., and A.Z.B. prepared the samples. A.H.B.A., J.Z., and I.S. performed the device measurements. A.H.B.A., J.Z., and J.D.S.Y. analyzed the experimental data. E.B.B. and T.S. developed the theoretical model and performed the theoretical calculations. T.T. and K.W. grew and provided the hBN crystals. A.H.B.A., J.Z., I.S., A.Z.B., E.B.B., T.S., and J.D.S.Y. discussed the findings and wrote the paper. J.D.S.Y supervised the project.

## Competing interests

The authors declare no competing interests.
