## [Peer Review File · Nature Communications]

REVIEWER COMMENTS

Reviewer #1 (Remarks to the Author):

In this work, the authors study high-bias electron transport in high-quality graphene devices. Their findings reveal that when the drift velocity of electron flow surpasses the speed of sound, the resistivity of graphene exhibits a dependence on the distance between the voltage probes and the current injection electrode. This dependence can be attributed to the Cherenkov-like emission of acoustic phonons, triggered by the out-of-equilibrium distribution of the electron flow. The population of phonons increases with distance from the current injector, resulting in a spatial variation in the measured resistance. The authors provide solid support for this explanation through their theoretical modeling.

The paper is well written, and the results are both novel and interesting. It should be published in Nature Communications if the authors address the following questions:

1. To ensure their results aren't influenced by high-bias induced self-gating, the authors should provide additional data and arguments. They briefly discuss this in the methods section, but it would be more convincing to present the measured two-probe voltage drop (e.g., for the curves in Fig. 1c) and compare it with the gate voltage.

Additionally, they need to show alternative methods for reversing the direction of current, such as swapping source and drain electrodes or testing the Onsager relation by switching current and voltage probes.

2. The paper's title, abstract and introduction emphasize THz frequencies, but the relationship of the results to this frequency range is not immediately clear. The authors should further explain how their findings are connected to these frequencies, particularly considering that their modeling indicates phonons with a 40GHz frequency (Fig. S11). In the last paragraph (line 274-275) they also mention that they apparently get phonons with the frequency ranging between 0.3-2.2 THz. It would be helpful if they could explain their estimations, for example in Supplementary information or methods.

3. In Fig. 2b, the curves at lower carrier density seem to exhibit growth only at relatively high drift velocities compared to the curves at high carrier density. The authors should provide comments about the reasons for this difference in behavior.

4. Fig. 2d displays curves with a non-monotonic dependence on carrier density. The authors should explain the origin of this behavior and provide details on how these curves were measured.

5. In Fig. 2c and 2d, the authors present data as a function of carrier density which also shows the neutrality point for various drift velocities. They need to clarify how they calculated the drift velocity at zero carrier density, as it appears that their formula gives an infinite drift velocity for any applied current.

6. Despite the expected self-gating contribution near the neutrality point, the NPs at different drift velocities seem to show consistent positions. The authors should explain this observation.

7. In line 69, the authors state that away from the Dirac point, interband transitions and optical phonon generation are suppressed. This is not accurate for optical phonons, as they cannot be generated at low carrier density when the Fermi energy is less than the energy of the optical phonon, see Ref 25. The authors should revise their statement accordingly.

8. In light of the comparison with reference 25, authors should provide comments on why, despite studying similar devices and a similar carrier density range, their conclusions appear to differ.

9. Authors probably missed another recent relevant work, that discusses high-bias induced phonon emission in high-quality graphene devices, see Nature Communications volume 12, Article number: 6392 (2021).

Reviewer #2 (Remarks to the Author):

Aaron H. Barajas-Aguilar and colleagues have demonstrated electrical amplification of acoustic phonons in graphene, revealing that such amplification can notably alter the electrical behavior of graphene-based devices. They observed a super-linear increase in resistivity parallel to carrier flow, a significant discovery that builds upon previous evidence of acoustic phonon amplification in graphene detected via noise measurements. However, the impact of terahertz (THz) acoustic waves on electronic properties was shown through transport measurement techniques, a novel approach detailed in their paper.

While the findings are compelling, I have several concerns about the present draft:

1. Figures 2c, 2d, and S4 show curves that appear abruptly truncated at higher values of drift velocity, v_d . Could the authors elaborate on this apparent artificial truncation?

2. The absence of this behavior in devices with a moiré superlattice raises questions. The satellite peak is positioned at a carrier density ($\sim 2.2 \times 10^{12} \text{ cm}^{-2}$) that far exceeds the characteristic carrier density discussed in the main text. If the carrier density falls below this threshold, the band structure of graphene aligned with hBN resembles that of the nonaligned structure. Thus, the rationale presented in the paper does not fully convince me.

3. Figure S6d shows asymmetric behavior in resistivity concerning the distance from the source electrode upon reversing carrier flow, deviating from what is observed in Device A. Specifically, resistivity decreases

with distance from the launching position, which is counterintuitive. Can the authors provide further commentary on this?

4. Additionally, Figure S7d lacks a peak in the differential voltage/current (dV/dI) around the carrier density of $n \sim 2 \times 10^{12} \text{ cm}^{-2}$, and there's no observed downturn at higher electron doping levels.

5. Lastly, on line 247, the equation $\hbar v_s 2k_F = 40 \text{ K}$ appears dimensionally inconsistent. The terms on each side of the equation do not seem to match in dimensions. Perhaps there's a more accurate expression that can be used.

In summary, while the study presents fascinating insights into phonon-electron interactions in graphene, the manuscript would benefit from a clearer exposition of these experimental observations and a more rigorous treatment of the data presented.

Reviewer #3 (Remarks to the Author):

The work Electrically-driven amplification of terahertz acoustic waves in graphene, by Barajas-Aguila and collaborators, reports on experiments showing directional growth of graphene resistance

induced by electrically-amplified acoustic phonons. Important applications may result from the effect addressed by the authors, such as electrical generation of terahertz acoustic phonons, as well as other types of condensed matter excitations such as terahertz electromagnetic source, and acoustic plasmons and magnons. In my evaluation, experimental results are quite sound and the analysis of the physical mechanisms behind the modulation of the resistivity across the sample and the amplification of acoustic phonon modes is very thorough. Authors also address other possible sources of the experimental results and argue convincingly that they are not operative under the experimental conditions in their study. I find that the work is significant to the field of transport in graphene, and also as a possible mechanism for generation of the aforementioned excitations in a graphene system. In my evaluation, the work meets the criteria of relevance and novelty of Nature Communication, and I recommend publication of the work in its present form.

Reply to reviewer comments

Reviewer #1 (Remarks to the Author):

In this work, the authors study high-bias electron transport in high-quality graphene devices. Their findings reveal that when the drift velocity of electron flow surpasses the speed of sound, the resistivity of graphene exhibits a dependence on the distance between the voltage probes and the current injection electrode. This dependence can be attributed to the Cherenkov-like emission of acoustic phonons, triggered by the out-of-equilibrium distribution of the electron flow. The population of phonons increases with distance from the current injector, resulting in a spatial variation in the measured resistance. The authors provide solid support for this explanation through their theoretical modeling.

The paper is well written, and the results are both novel and interesting. It should be published in Nature Communications if the authors address the following questions:

Reply: We thank the reviewer for their careful evaluation of the manuscript as well as their feedback.

1. To ensure their results aren't influenced by high-bias induced self-gating, the authors should provide additional data and arguments. They briefly discuss this in the methods section, but it would be more convincing to present the measured two-probe voltage drop (e.g., for the curves in Fig. 1c) and compare it with the gate voltage.

Reply: Following the reviewer's request, we have added a new SI Figure S7 that shows the applied source-drain bias V_{SD} vs. current and the two-probe differential resistivity vs current at different gate voltages (V_G). It shows how V_{SD} never exceeds 0.6 V, and is small compared to the applied gate voltages where phonon amplification is clearly observed (6 to 50 V). We address this point further in the response to comment #6.

Figure-S7. Two probe measurements for device A. Applied source drain voltage vs. current (top) and two probe measurements of the differential resistivity vs. current (bottom) at different carrier densities. The curves appear symmetric, as the spatial growth of the resistance is hidden in the total two-probe resistance of the device.

“Additionally, they need to show alternative methods for reversing the direction of current, such as swapping source and drain electrodes or testing the Onsager relation by switching current and voltage probes.”

Reply: We have added new data to the SI showing swapped source and drain electrodes for Device B. Figure S8c now shows both measurement configurations. As expected, the data is invariant to reversing the sign of the bias while swapping the source and drain electrodes. We do not have the same data for Device A, but we note that the measurements are DC, so swapping the source and drain electrodes amounts to an overall relative voltage shift which does not change the physics (Ex: $V_{\text{source}} = 0.6 \text{ V}$, $V_{\text{drain}} = 0$, $V_{\text{gate}} = 10 \text{ V}$ would map to $V_{\text{source}} = 0$, $V_{\text{drain}} = -0.6 \text{ V}$, $V_{\text{gate}} = 9.4 \text{ V}$).

Figure-S8. “c) Raw V-I curves (top) and differential resistivity vs. source-drain current (bottom) for all the pairs of contacts in device B at $n = -1.4 \times 10^{12} \text{ cm}^{-2}$ (hole doped, $V_g = -21 \text{ V}$). The lighter thick lines correspond to the forward current (from source to drain) while the solid thin lines are for the case of reverse current, where the source and drain cables are swapped.

2. The paper's title, abstract and introduction emphasize THz frequencies, but the relationship of the results to this frequency range is not immediately clear. The authors should further explain how their findings are connected to these frequencies, particularly considering that their modeling indicates phonons with a 40GHz frequency (Fig. S11). In the last paragraph (line 274-275) they also mention that they apparently get phonons with the frequency ranging between 0.3-2.2 THz. It would be helpful if they could explain their estimations, for example in Supplementary information or methods.

Reply: The key point is that the amplified phonons have relatively large wavevectors, comparable to the Fermi wavevector, which allows them to strongly scatter the electrons. Such phonons will be in the THz frequency range. If the phonons were of lower wavevector/frequency, they would not be able to scatter the electrons effectively. This is consistent with our experimental observations and supported by the theoretical analysis.

As mentioned in the main text (Figure 1 caption, lines 89-90), the frequency of the amplified phonons can be estimated as $E = \hbar v_{\text{sound}} * 2k_F$, where $k_F = \sqrt{\pi n}$. A plot of this curve for the longitudinal and transversal phonons is included below. The minimum frequency reported (0.3 THz) corresponds to that of the transversal phonons at $0.4 \times 10^{12} \text{ cm}^{-2}$, the lowest carrier density where we measure phonon amplification. The maximum frequency (2.2 THz) corresponds to that of the longitudinal phonons at the highest carrier density reached in our measurements. We have added this explanation and plot to the SI section 1.5 to make this more clear.

“their modeling indicates phonons with a 40 GHz frequency” -

Reply: The 40 GHz values in Figure S13 (previously S11) are the predicted **rates** of phonon emission by the electrons, which is different from the THz phonon frequencies.

3. In Fig. 2b, the curves at lower carrier density seem to exhibit growth only at relatively high drift velocities compared to the curves at high carrier density. The authors should provide comments about the reasons for this difference in behavior.

Reply: The resistance growth is determined by the net rate of phonon amplification, which grows with the electron-phonon coupling and the drift velocity. Hence, a larger drift velocity can compensate for other effects that would reduce the amplification rate, such as a reduction in electron-phonon coupling. At lower carrier densities, the electron-phonon coupling is reduced, leading to a reduced amplification at a constant drift velocity. This is observed in Fig. 2d where the differential resistance decreases with carrier density when biasing at a constant high drift

velocity. This reduction can be partially compensated by biasing to higher drift velocities, which increase the overall amplification rate to create an observable resistance growth.

In the main text we provide an additional explanation in lines 237-238: “This explains why at lower carrier densities, higher drift velocities are required to observe the phonon amplification effect (Fig. 2b inset).”

4. Fig. 2d displays curves with a non-monotonic dependence on carrier density. The authors should explain the origin of this behavior and provide details on how these curves were measured.

Reply: As explained in the discussion section (lines 232 to 246), the non-monotonic dependence arises from the competition between phonon emission and phonon decay, which determine the net amplification rate as $\Gamma_{amp} = \Gamma_{emission} - \Gamma_{decay}$. In short, $\Gamma_{emission}$ causes the initial increase with n , but is eventually overtaken by Γ_{decay} at high n , leading to a peak in Γ_{amp} .

$\Gamma_{emission}$ increases with n due to increased electron-phonon coupling and the larger phase space for transitions. This causes the initial increase observed in Fig 2d for $|n| > 0.4 \times 10^{12} \text{ cm}^{-2}$, and is supported by our theoretical calculations in Fig. S13 which predicts a linear dependence of amplification rate on carrier density.

Γ_{decay} also increases with n because the frequency of emitted phonons increases as $\omega \propto k_F \propto \sqrt{n}$, and higher-frequency phonons have higher rates of anharmonic decay from phonon-phonon interactions. Anharmonic decay will typically have a ω^3 or higher power dependence, so Γ_{decay} will initially grow slowly with n , but then will eventually overtake $\Gamma_{emission}$ at higher carrier densities, leading to a suppression of the resistance growth effect.

Note, as explained in our methods section, we do not include the effects of anharmonic decay in our model, so this downturn does not appear in the theoretical analysis.

Note, that the location of the peak effect is non-intrinsic and is likely sample-dependent. For example, disorder and strain can affect the emission and decay rates in different ways that are sample dependent.

“provide details on how these curves were measured.” (referring to Figure 2d)

The data presented in Figures 1 and 2 come from the same measurement dataset following the methods described in the main text. We sweep a DC source-drain voltage while measuring the source current and the voltage difference between consecutive electrodes as a function of gate voltage. From these measurements, we calculate the local resistivity and local differential resistivity between adjacent contacts. We also can calculate the carrier density n ($n = CV_G/e$) and drift velocity for each datapoint as $v_D = j/ne$. This allows us to replot the dataset for constant values of v_D to produce the plots in Figure 2b,c,d. Figure 2d is a plot of the differential resistivity versus carrier density while holding v_D constant.

Note that v_D depends on the sign of the current and the carrier density, so for Figure 2c&d the current bias switches sign as the carrier density goes from negative to positive.

We have added a detailed explanation of the data analysis in the methods section.

5. In Fig. 2c and 2d, the authors present data as a function of carrier density which also shows the neutrality point for various drift velocities. They need to clarify how they calculated the drift velocity at zero carrier density, as it appears that their formula gives an infinite drift velocity for any applied current.

Reply: We thank the reviewer for bringing up this point. The curves in Figures 2c&d are of constant drift velocity $v_D=j/ne$. None of our data points are strictly at $n=0$, the smallest n value is $6 \times 10^{10} \text{ cm}^{-2}$, so the drift velocity calculation is always well-defined.

6. Despite the expected self-gating contribution near the neutrality point, the NPs at different drift velocities seem to show consistent positions. The authors should explain this observation.

Reply: Indeed, self-gating effects can play a larger role near the Dirac point, but these do not appear in Figure 2c&d because the curves are lines of constant drift velocity ($v_D=j/ne$), as opposed to curves of constant bias voltage V_{SD} . As such, as the magnitude of carrier density decreases, the current density and V_{SD} also decreases to keep v_D constant. This is clearly shown in the below figure, where we plot the corresponding V_{SD} for each curve in Fig 2c&d. At $n=6 \times 10^{10} \text{ cm}^{-2}$, which is the lowest n value reached for $v_D=200 \text{ km/s}$, the applied V_{SD} is 155 mV while the gate voltage is 2 V. For this reason, self-gating is a very small effect on the location of the Dirac peaks in these curves

Figures 2c&d from main text

V_{SD} vs. n for figures 2c&d from main text

7. In line 69, the authors state that away from the Dirac point, interband transitions and optical phonon generation are suppressed. This is not accurate for optical phonons, as they cannot be generated at low carrier density when the Fermi energy is less than the energy of the optical phonon, see Ref 25. The authors should revise their statement accordingly.

Reply: We have reworded the main text to be more clear as follows: "we measure the position-dependent resistance, focus on gate voltages away from the Dirac point so interband transitions are suppressed, and apply moderate source-drain voltage biases to avoid optical phonon generation"

8. In light of the comparison with reference 25, authors should provide comments on why, despite studying similar devices and a similar carrier density range, their conclusions appear to differ.

Reply: The key difference between our work and reference 25 are the following:

- (1) **We study much longer graphene devices** ($13\ \mu\text{m}$ and $14.5\ \mu\text{m}$ for our Devices A and B, respectively, compared to the $\sim 4\ \mu\text{m}$ long device in reference 25). This is important because the phonon amplification leads to a superlinear growth of the resistance with distance, making it harder to observe in a $4\ \mu\text{m}$ long device.
- (2) **We study the spatial dependence of the resistivity** using multiple voltage terminals, which allows us to observe the resistance growing in the direction of carrier flow.
- (3) **Our observed effects occur at relatively low electric fields.** The largest electric field we measured in our study is $0.07\ \text{V}/\mu\text{m}$. We specifically limited the applied bias to avoid optical phonon generation. Reference 25 applies nearly 10x larger electric fields up to $0.5\ \text{V}/\mu\text{m}$. Due to the large electric fields and drift velocity, reference 25 interprets their nonlinear effects as arising from optical phonon emission in the hBN, which we rule out in our manuscript due to the low drift velocities at which we observe the phonon amplification effects.

9. Authors probably missed another recent relevant work, that discusses high-bias induced phonon emission in high-quality graphene devices, see Nature Communications volume 12, Article number: 6392 (2021).

Reply: We thank the reviewer for the reference. This work is focused on studying high-bias effects in graphene devices in a magnetic field, where different resistance features are observed and ascribed to phonon emission between cyclotron orbit states. Interestingly, they observe a resistance peak at high magnetic fields when the drift velocity is close to the transverse sound velocity, although the measurements are performed at 40K, and no carrier density or length dependence is presented. They only explore small drift velocities, at most 1.1x the sound velocity.

This study is very different from ours, where we observe a non-linear growth of the resistance at zero magnetic field that is activated when the drift velocity exceeds the sound velocity. Different physics are at play under the quantizing effect of the high magnetic fields.

We have included a citation in the main text to this paper as an example of a previous graphene work at high bias (now reference 28).

Reviewer #2 (Remarks to the Author):

“Aaron H. Barajas-Aguilar and colleagues have demonstrated electrical amplification of acoustic phonons in graphene, revealing that such amplification can notably alter the electrical behavior of graphene-based devices. They observed a super-linear increase in resistivity parallel to carrier flow, a significant discovery that builds upon previous evidence of acoustic phonon amplification in graphene detected via noise measurements. However, the impact of terahertz (THz) acoustic waves on electronic properties was shown through transport measurement techniques, a novel approach detailed in their paper.

While the findings are compelling, I have several concerns about the present draft:

Reply: We thank the reviewer for their evaluation of the manuscript and feedback. We reply to their concerns below.

“1. Figures 2c, 2d, and S4 show curves that appear abruptly truncated at higher values of drift velocity, v_d . Could the authors elaborate on this apparent artificial truncation?”

Reply: The curves in Figure 2 appear truncated because the measurements are taken within a fixed point grid corresponding to a source-drain voltage range of $V_{SD} = -0.6$ to 0.6 V. These voltage limits are set to avoid heating and possible device damage. When we plot this dataset of constant V_{SD} range along different axes, such as current or drift velocity, the curves will appear as different ranges because $I = V_{SD}/R$ and $v_d = V_{SD}/(RWne)$, where I is current, R is resistance, W is sample width, and n is carrier density.

Likewise, in Figure 2c&d we plot curves of constant v_d , and so the range of carrier densities n where we have data are restricted as $n = V_{SD}/(RWv_d e)$, where V_{SD} is limited to ± 0.6 V. From this equation, we see that the range of n values is especially restricted at large drift velocities.

To make this aspect of the plots more clear, we have added the following explanation to the figure caption:

“Note that data was taken with a maximum source-drain voltage of ± 0.6 V, hence the curves appear with different ranges when plotted versus current, v_d , or for constant values of v_d .”

2. The absence of this behavior in devices with a moiré superlattice raises questions. The satellite peak is positioned at a carrier density ($\sim 2.2 \times 10^{12} \text{ cm}^{-2}$) that far exceeds the characteristic carrier density discussed in the main text. If the carrier density falls below this threshold, the band structure of graphene aligned with hBN resembles that of the nonaligned structure. Thus, the rationale presented in the paper does not fully convince me.

Reply: We agree that the electronic dispersion can be nearly linear away from the satellite peaks. However, the large Moire pattern results in a substantially smaller Brillouin zone that is comparable to the magnitudes of the wavevectors for the electrons and emitted phonons (Note $k_{\text{phonon}} \sim k_{\text{Fermi}}$). For example, a carrier density of $1 \times 10^{12} \text{ cm}^{-2}$ corresponds to a Fermi wavevector of $1.7 \times 10^8 \text{ m}^{-1}$, which is 65% of the distance to the edge of the superlattice Brillouin zone for a 14 nm moire. As a result, Umklapp scattering is unlocked, as electron-electron and phonon-phonon scattering can result in wavevectors that fall outside the first Brillouin zone. Umklapp scattering will greatly increase phonon-phonon scattering and decrease the phonon lifetime which is key to the amplification effect.

In principle, Umklapp scattering can be suppressed at very low carrier densities, but then phonon amplification is also suppressed as well.

In normal unaligned graphene devices, Umklapp processes are forbidden because the Brillouin zone is at least 100x larger than the typical wavevectors of electrons and emitted phonons.

The effects of Umklapp scattering in aligned-graphene hBN devices have been clearly demonstrated in Wallbank et. al. Nat Phys 2019 (Ref 34), where it results in an enhanced resistivity due to increased electron-electron scattering. This Umklapp-induced resistance is observed over a wide range of carrier densities, even far away from the superlattice Dirac points. We observe this in aligned device C, where the overall resistivity of the device is 50 times higher than in the unaligned devices A&B. This increased resistivity can additionally suppress phonon amplification, as it means that larger power dissipation is required to reach high drift velocities, which will cause deleterious heating

We have modified the explanation in the main text (lines 258-262) to more explicitly connect with Umklapp scattering as follows:

“In the case of aligned graphene-hBN devices, the moiré superlattice can be as large as 14 nm, which is comparable to the wavelengths of the emitted phonons (9 to 40 nm). As such, the superlattice will induce phonon-phonon Umklapp scattering, which will reduce the phonon lifetime and the amplification effect (ref 34).”

“3. Figure S6d shows asymmetric behavior in resistivity concerning the distance from the source electrode upon reversing carrier flow, deviating from what is observed in Device A. Specifically, resistivity decreases with distance from the launching position, which is counterintuitive. Can the authors provide further commentary on this?”

Reply: In Device B, we observe the resistivity to grow in the direction of carrier flow, consistent with Device A, but the growth is larger for right-moving carriers as compared to left-moving (Figure S8d, previously S6d). We address this point in the SI by noting that Device B has spatially inhomogeneous disorder. This is observed as a local resistivity that increases across the channel under zero/low bias conditions from 4 Ohms near the source to 22 Ohms near the drain). Therefore, Device B’s resistance profile includes the combined effects of both the phonon growth and the disorder; inverting carrier direction flips the phonon growth profile, but has no effect on the disorder profile. For this reason, right-moving carriers show a larger resistance growth than left-moving carriers, consistent with both the phonon growth model and the pattern of disorder.

By contrast, Device A has a uniform resistivity that varies by only 6% across the channel under zero bias conditions, and the right-moving and left-moving carriers have similar spatial profiles.

We have modified the supplementary information so that the explanation of the Device B data is clearer.

4. Additionally, Figure S7d lacks a peak in the differential voltage/current (dV/dI) around the carrier density of $n \sim 2 \times 10^{12} \text{ cm}^{-2}$, and there's no observed downturn at higher electron doping levels.

Reply: We assume that Reviewer 2 is referring to Figure S9c (previously S7c) in comparison to Figure 2d, which are differential resistance vs n curves at constant v_{drift} for Device B and Device A, respectively. In Device A, we observe a clear peak at $n = 2 \times 10^{12} \text{ cm}^{-2}$, while in Device B we potentially observe a peak developing, but shifted to higher values of $|n| \sim 3 \times 10^{12} \text{ cm}^{-2}$.

Based on our observations and theoretical model, the carrier density where this peak occurs is not intrinsically special and likely dependent on extrinsic factors such as disorder or strain, which will vary between devices. Even for Device A, there are differences between the electron and hole sides of the n dependence in Figure 2d - the hole side grows faster with $|n|$. Moreover, the location of the peak shifts with v_{drift} .

A full quantitative model of the carrier density dependence of phonon amplification is beyond the scope of this paper. But, within our model, the presence of a peak can be explained. We repeat the explanation given above for reviewer 1 comment #4:

The phonon amplification effect is determined by the net amplification rate:

$\Gamma_{amp} = \Gamma_{emission} - \Gamma_{decay}$, where $\Gamma_{emission}$ and Γ_{decay} are the phonon emission and decay rates, respectively.

Both $\Gamma_{emission}$ and Γ_{decay} are expected to grow with n . A peak in Γ_{amp} will occur if $\Gamma_{emission}$ grows faster at first, but then is eventually taken over by Γ_{decay} at higher n values.

$\Gamma_{emission}$ increases with n due to increased electron-phonon coupling and the larger phase space for transitions at higher carrier densities. Γ_{decay} will grow with n because the frequency of emitted phonons increases as $\omega \propto k_F \propto \sqrt{n}$, and higher-frequency phonons have higher rates of anharmonic decay from phonon-phonon interactions. Anharmonic decay will typically have a ω^3 or higher power dependence, so Γ_{decay} will initially grow slowly with n , but then will eventually overtake $\Gamma_{emission}$ at higher carrier densities, leading to a suppression of the resistance growth effect.

Both the emission and decay rates will be affected differently by disorder, and hence the location of the peak is expected to change due to sample-to-sample variations in these quantities.

5. Lastly, on line 247, the equation $\hbar v_s 2k_F = 40 \text{ K}$ appears dimensionally inconsistent. The terms on each side of the equation do not seem to match in dimensions. Perhaps there's a more accurate expression that can be used.

Reply: We thank the reviewer for catching this error. We replaced the mentioned equation with $\hbar v_s 2k_F / k_B = 40 \text{ K}$ (where k_B is Boltzmann's constant).

In summary, while the study presents fascinating insights into phonon-electron interactions in graphene, the manuscript would benefit from a clearer exposition of these experimental observations and a more rigorous treatment of the data presented.

Reviewer #3 (Remarks to the Author):

"The work Electrically-driven amplification of terahertz acoustic waves in graphene, by Barajas-Aguila and collaborators, reports on experiments showing directional growth of graphene resistance induced by electrically-amplified acoustic phonons. Important applications may result from the effect addressed by the authors, such as electrical generation of terahertz acoustic phonons, as well as other types of condensed matter excitations such as terahertz electromagnetic source, and acoustic plasmons and magnons. In my evaluation, experimental results are quite sound and the analysis of the physical mechanisms behind the modulation of the resistivity across the sample and the amplification of acoustic phonon modes is very thorough. Authors also address other possible sources of the experimental results and argue convincingly that they are not operative under the experimental conditions in their study. I find that the work is significant to the field of transport in graphene, and also as a possible mechanism for generation of the aforementioned excitations in a graphene system. In my evaluation, the work meets the criteria of relevance and novelty of Nature Communication, and I recommend publication of the work in its present form."

Reply: We thank the reviewers for their evaluation of our manuscript.

—

Other changes:

In the new submission, we have made a small change to how curves are calculated for Figure 2c&d. The curves are effectively identical as before, except at the Dirac point where the values change by ~ 10% (see comparison below). None of our analyses or conclusions are affected by this change.

For transparency, the change is to use interpolation to determine the points, which is necessary as we plot equally-spaced values of constant v_D , but the original dataset was taken with points of equally-spaced voltage bias. This is only important near the Dirac point, where the distance between v_D data points are more spread out due to the low carrier density. This interpolation is

valid as the differential resistance versus drift velocity curves are smooth functions (see Figure 2b). A description of this process is described under “Data Analysis” in the Methods section.

Note, this correction only affects Figures 2c&d, as well as other figures which take constant v_D cuts of the data (Figures S4 and S9 in SI).

Sample of Figure 2d data showing a comparison between the original curves and the new curves in the resubmission where we use interpolation to take constant v_D linecuts.

Dark thin lines = new, with interpolation

light wide lines = original, no interpolation

Curves are offset horizontally for clarity.

REVIEWERS' COMMENTS

Reviewer #1 (Remarks to the Author):

The authors have addressed my comments, and I recommend this work for publication in Nature Communications.

Reviewer #2 (Remarks to the Author):

All concerns have been thoroughly addressed, and I have no further questions at this time. It can be accepted in its current form.